# Effect of Digital Game-Based Learning on Student Engagement and Motivation

Muhammad Nadeem *, Melinda Oroszlanyova and Wael Farag

College of Engineering and Technology, American University of the Middle East, Egaila 54200, Kuwait; melinda.oroszlanyova@aum.edu.kw (M.O.); wael.farag@aum.edu.kw (W.F.)
* Correspondence: muhammad.nadeem@aum.edu.kw

**Abstract:** Currently, academia is grappling with a significant problem—a lack of engagement. Humankind has gone too far into exploring entertainment options, while the education system has not really kept up. Millennials love playing games, and this addiction can be used to engage and motivate them in the learning process. This study examines the effect of digital game-based learning on student engagement and motivation levels and the gender differences in online learning settings. This study was conducted in two distinct phases. A game-based and traditional online quizzing tools were used to compare levels of engagement and motivation, as well as to assess the additional parameter of gender difference. During the first phase of the study, 276 male and female undergraduate students were recruited from Sophomore Seminar classes, and 101 participated in the survey, of which 83 were male and 18 were female. In the second phase, 126 participants were recruited, of which 107 (63 females and 44 males) participated in the anonymous feedback surveys. The results revealed that digital game-based learning has a more positive impact on student engagement and motivation compared to traditional online activities. The incorporation of a leaderboard as a gaming element in the study was found to positively impact the academic performance of certain students, but it could also demotivate some students. Furthermore, female students generally showed a slightly higher level of enjoyment toward the games compared to male students, but they did not prefer a comparison with other students as much as male students did. The favorable response from students toward digital game-based activities indicates that enhancing instruction with such activities will not only make learning an enjoyable experience for learners but also enhance their engagement.

**Keywords:** gamification; digital game-based learning; engagement; pedagogy design; motivation; leaderboard; emerging technologies



## 1. Introduction

The world of academia is currently facing a serious crisis—a crisis of engagement [1–4]. Humankind has gone too far into exploring entertainment options, while the education system has not really kept up and has barely progressed beyond 1900. The traditional methods used for engagement no longer hold our attention the way they once did because we have become used to so much engagement per second. This situation is exacerbated when teaching online or in large classes at universities, where there is little or no interaction between students and teachers. COVID-19 compelled the academic sector to embrace online learning as a means of survival [4]. This abrupt shift left instructors stranded on how to engage students in this mode of setting and facilitate learning. Many instructors faced difficulty in engaging students in a meaningful way during the online delivery [5]. This might be due to distractions, lack of supervision, and nonacademic use of digital devices mainly for interacting with social media and playing games. Furthermore, the lack of learning of interaction with the teacher and in-person loss of learning is sometimes critical for a student in need of assistance. Strategies such as scaffolding the learning material, synchronous class meetings, use of breakout rooms, use of quizzing programs,

and polling helped overcome the pyramid of challenges posed by remote teaching [6]. The implementation of these strategies necessitates the employment of technology; fortunately, numerous applications are at our disposal. The integration of online activities has been highly effective in fostering student engagement during in-person teaching, as this approach was previously uncommon for students and succeeded in capturing their attention [3,7,8]. Initially, online activities were instrumental in engaging students during the COVID-19 pandemic owing to their innovative nature. However, their appeal gradually diminished as these methods became routine, resulting in a decline in motivation and making it challenging for students to concentrate on course content, whether in online or in-person classes. Therefore, it is crucial to explore alternative digital technologies, such as AI-generated applications [9], chatbots [10], podcasting [11], interactive whiteboards, augmented reality [12,13], and gamification [14]. Many of these technologies have been explored extensively, while others are emerging and not quite mature yet. AR and gamification are two relatively new technologies that have the potential to attract and engage students, thereby promoting active learning. The deployment of AI-generated applications has many ethical issues that need to be solved before they can be used freely in class. The value of interpersonal communication and human engagement in the classroom might be diminished by the usage of chatbots. Podcasting is not an active method, and interactive whiteboards are not only expensive, but also fragile, and their use is problematic in large classrooms, especially for backbenchers. AR is a relatively new technology, but it has its own issues which make it hard to implement in the classroom. It requires a device with higher computation power, a high-speed internet connection, long battery life, and greater data storage capacity. Furthermore, there is a lack of tools or apps that can quickly be adopted for instruction design. On the other hand, with the increasing popularity of video games across all age groups, digital game-based learning can prove to be a valuable tool in energizing students and encouraging active learning in the classroom. Game-based or gamified activities have been utilized for a considerable period to create courses that not only make learning enjoyable but also engage and motivate students in both traditional and online learning environments [15]. Furthermore, playing games also improves decision-making, and those who play action-based video games make judgments 25% faster than the average person while maintaining the same level of accuracy. The gamers make decisions and take actions up to six times per second, which is four times quicker than the average person [16]. In the future, the immersive and interactive elements of gamification may be improved with the help of AR and VR technologies, which also opens up new opportunities and problems. We believe that augmented reality game-based learning (ARGBL) will become increasingly relevant in technology-enhanced learning as the above-mentioned issues are resolved.

Digital game-based learning involves linking subject content to gameplay, enabling students to apply their knowledge in real-world scenarios, thereby making learning more engaging and enjoyable. Through game-based activities, students are presented with challenges and obstacles that motivate them to work harder, increasing their sense of accomplishment and self-esteem upon overcoming these obstacles. Additionally, games involve social interactions with other players, allowing students to establish relationships and a sense of community, which can be crucial in online learning environments where students may feel isolated [17]. Most importantly, games are entertaining, and, when students enjoy themselves, they are more likely to be motivated to continue learning. Overall, game-based activities can be a highly effective way to engage and motivate students, enabling them to invest more in their learning and achieve better outcomes.

The incorporation of game-like characteristics into a non-game environment is referred to as gamification. The most commonly used gamification elements are points, badges, and leaderboards. Leaderboards are a vital component of game design and a powerful tool for motivating users through competition [18]. However, certain studies suggest that the use of leaderboards can demotivate students [19,20].

Game-based learning often requires specialized hardware or software, which can be expensive and time-consuming to set up and maintain. Technical issues such as glitches or system crashes can also interfere with learning and frustrate students [21]. If it is too tedious or expensive, then the instructor might not be motivated to implement them in the class. Therefore, instructors have to analyze factors such as cost, developing tasks, and availability of digital resources before using games in the classroom. In addition, games should be compatible with the course curriculum and learning goals. The games that educators choose to use must be pertinent and successful in reaching learning objectives [22]. Considering these arguments, we enhanced the online teaching instruction with simple game-based activities implementing sorting games such as Vortex for exploring student engagement and motivation. Furthermore, gender is one of the many factors that can influence an individual's perception of game-based learning, but it is not the only factor. Research has shown that both males and females can have positive perceptions of game-based learning, but the specific games and learning outcomes that appeal to each gender may vary. Therefore, it is important to explore the differences in gender perception toward Vortex-based quizzes.

To sum up, research has shown that, when students have ownership of an educational tool, it can enhance their attitude, motivation, interest, and learning outcomes [23]. Many online activities and games provide opportunities for students to take control, which is why they often return to these games to practice. Additionally, these activities require minimal teacher intervention, as the game itself usually provides answers or results. This promotes students' self-reliance when using digital resources and minimizes disruptions to their work.

### 1.1. Research Question

The goal of this study is to ascertain how using Vortex as a game in a non-game learning setting has an effect from the viewpoints of university undergrads. Specifically, it examines student motivation and engagement levels, as well as gender differences, when using a digital game-based quiz tool and a conventional tool. The study was conducted in two phases, where two different digital platforms were used in the first phase to compare engagement and motivation as well as the impact of the leaderboard. We used a game-based platform only during the second phase to examine gender differences. Inspired by Keller's motivational design process, the lecture component of an online course was redesigned to include game-based learning activities. Students can complete these activities in class using personal devices. The students were provided time to complete the activity at the beginning, middle, and end of the lecture. This study answers the following research questions:

- RQ 1: How are students' engagement and motivation affected by the usage of game-based activities?
- RQ 2: What is the impact of the leaderboard on student motivation and behavior?
- RQ 3: Do males or females have the same perception toward game-based activities?

### 1.2. Literature Review

More than 2.6 billion people enjoy video or computer games worldwide, and the number is growing daily. For instance, according to the NPD Group (NPD, 2019), the number of individuals playing computer games increased by 6% between 2018 and 2019. Every other person aged 6 to 64 who lived in the five main European nations of France, Italy, Germany, Spain, and the UK played video games, according to statistics gathered by the Interactive Software Federation of Europe [24]. The information showed that players of all ages enjoyed playing games on the computer, these are especially popular among young people. Among young poll participants, the percentage of players reached 70% (15–25 years old). The typical player's weekly gaming time was estimated to be 9.5 h, and 47% of all players who took part in the survey identified as female. The Statistical Office of the European Union [25] carried out a larger poll in 2019 that covered the majority of

European nations. According to this survey's data, which was gathered in 2018, 33% of all individuals between the ages of 16 and 74 had played video games.

Several educational and psychological theories support the use of game-based learning. Constructivism theory suggests that students learn the best when they actively construct their knowledge through experiences and interactions [26]. Games provide an opportunity for students to explore, experiment, and make decisions that impact the outcomes of the game. This also helps them to construct their knowledge in a meaningful way. The experiential learning theory [27] emphasizes the importance of direct experience and reflection in the learning process. Games provide an opportunity for students to experience different scenarios and to reflect on their decisions, leading to a deeper understanding of the contents being learned. The flow theory [28] describes a state of deep engagement and enjoyment that can be achieved when a person is completely immersed in an activity. It suggests creating a balance between skill level and challenge to avoid anxiety and continue the flow of the game.

Many pieces of research have shown that most people enjoy playing games, and this enjoyment can lead to an increase in engagement and motivation during educational activities [29]. The concept of gamification has gained extensive attention in recent years as a possible way to make learning more enjoyable and engaging. Gamification involves incorporating game-like structures, such as points, badges, and leaderboards, into educational activities to increase motivation and engagement. Several studies in the academic literature have emphasized the advantages of incorporating gamification into the educational process, such as an enhancement of students' capacity to acquire new competencies, attendance, motivation, and participation in undergraduate course activities by using points, leaderboards, levels, and badges [30–33]. Although various gamification methods have been applied and implemented in educational contexts during the last decade to achieve certain goals, their effect on student performance and motivation is still questionable [18,19,34–36].

Many of the existing student response systems (SRSs) have incorporated game features to boost student engagement such as Space Race games in Socrative [37]. Kahoot! was the first to offer a gaming experience using game design concepts from the theory of intrinsic motivation [38] and game flow [39]. It has been used in a variety of ways to enhance teaching/learning in a fun and interactive way for engaging students and make learning more enjoyable and memorable [40–47]. Although these SRSs can be a useful tool for promoting engagement and assessing knowledge in the classroom, they cannot be used outside the classroom due to their synchronous nature as they always require a facilitator. In addition, they are time-consuming and have a limit on participants. Using such tools is fun but hampers the learning process. Therefore, they are suitable for occasional use but not as permanent instruction methods. The Vortex sorting game, on the other hand, is simple and has been used for improving vocabulary which is crucial for learning new topics [48].

A strong conclusion that arises from literature is the importance of the context. A contextualization strategy is generally thought to result in improved comprehension. Learners respond differently to the same gamification setup; effectiveness in one context may not be the same in another context. This vagueness about the use of gaming elements in educational settings makes it important to stress context-specific studies for a better understanding of its impact on student performance and motivation [35,49–52].

Leaderboards are the most commonly used method for course gamification [18]. Usefulness and relative easiness of setup are some of the apparent reasons for their frequent use. Leaderboards provide comparative feedback to the students of a gamified course about their performance against other students using points and ranks. Many studies have reported that leaderboards can positively impact the achievements of learners [18,53,54], engagement [30], and the amount of completed work [21,36,55], while maintaining performance [56] and course attendance [57]. While gamifying a course with leaderboards, the aforementioned favorable verdicts come with some concerns that need to be handled. These concerns can lead to affecting student participation, possibly even resulting in high drop-out rates if the course leaderboards

can only sustain engagement for a short time period [18,58]. If students focus on achieving the extrinsic goals offered by a leaderboard instead of the assigned tasks, this could distract the student learning and result in low-quality work [21,56,59].

The competitive nature of a leaderboard may attract higher-performing competitive students but can backfire with low-performing students [18,32,33,53,60]. Short-term performance gains from an extrinsically rewarding leaderboard may come at the cost of students' intrinsic motivation [19,56,58,60]. The authors in [19,56] reported a significant improvement in the learning performance of students in the gamified condition in programming and language courses. The latter also discovered that the students' focus on the extrinsic rewards used by the leaderboard uplifted their performance until the point they achieved the threshold and diminished consequently. They also pointed out that the use of points, rank, and forced social comparison to control behavior resulted in a shift from intrinsic to extrinsic motivation. However, the authors of [61] suggested otherwise, stating that incorporating competitive gaming elements, such as badges and leaderboards, may hurt low-performance students.

Students' personalities [62] and genders [22] affect their perception of game elements in an educational context, and gender disparities can impact students' academic performance. For instance, the authors of [63] discovered that female students excelled more than male students in a particular learning task. Likewise, the authors of [17] found that female students had higher academic achievements compared to male students. Despite both genders having comparable levels of intellectual capacity, this outcome was attributed to variations in their personality traits. It was shown by the authors of [64] that feedback might be more helpful to females than males, while less extroverted students may find progress bars more useful than those who are more extroverted and see the value in badges in gamified courses than their counterparts.

### 1.3. Vortex Sorting Game

Schwabe and Goth defined the criteria for creating gamifying instruction material to engage them in a meaningful way [65]. They underlined six structural elements that should characterize the game to achieve the goals in a fun way: rules, objectives and goals, outcome and feedback, competition and challenges, interaction, and story. We adhered to these criteria when selecting the game, called Vortex, for integration with teaching activities.

Vortex is the most recent game template released by ClassTools for designing online games, in which participants must classify words or phrases into up to four categories. When students play a Vortex game, they must sort terms that appear on the screen as quickly as possible. This game has very simple rules which make it very easy for students to follow without any cognitive overloading. The objective is to familiarize the students with the vocabulary of technical terms, definitions, and key points of the topic as soon as a topic is covered. The goal is to engage the students with the course material, and this retrieval-based learning strategy helps students learn faster. It can also work as a multiple-choice question or true–false quiz, forcing the student to choose the right option only. This also provides immediate feedback to students on their learning, as wrong answers are not accepted by the game, and the music generated indicates that the student's response is wrong. This type of confirmatory feedback reduces ambiguity about the concepts learned and helps in the attainment of goals. The scores are based on how quickly and accurately the terms are sorted. This also provides feedback on their retention of knowledge as a low score indicates that the student did not make many mistakes, indicating that the student has a good grasp of the topic. The student is not only competing against their previous efforts but also their fellow students as the scores are displayed on the leaderboard. This not only poses a challenge but also creates a competitive environment. This creates an opportunity to interact and opens a channel of communication with fellow students—another important element of course design.

Students can play Vortex game-based quiz activities (GBQAs) asynchronously during lectures, at the end of the day, near assessment times, and during their free time reviewing

the material. Moreover, Vortex is simple to use, develops activities easily and quickly, is free for instructors and students, is compatible with a wide range of devices, and is suitable for a variety of teaching and learning styles. Therefore, Vortex seems to offer great potential for involving and evaluating students while remaining simple to use, inexpensive, and adaptable. Digital devices such as smartphones, tablets, and laptops support Vortex-based formative assessment activities. These activities do not need the installation of any application and only require access to the Internet. A screenshot of the activity related to the topic of ethics is shown in Figure 1.

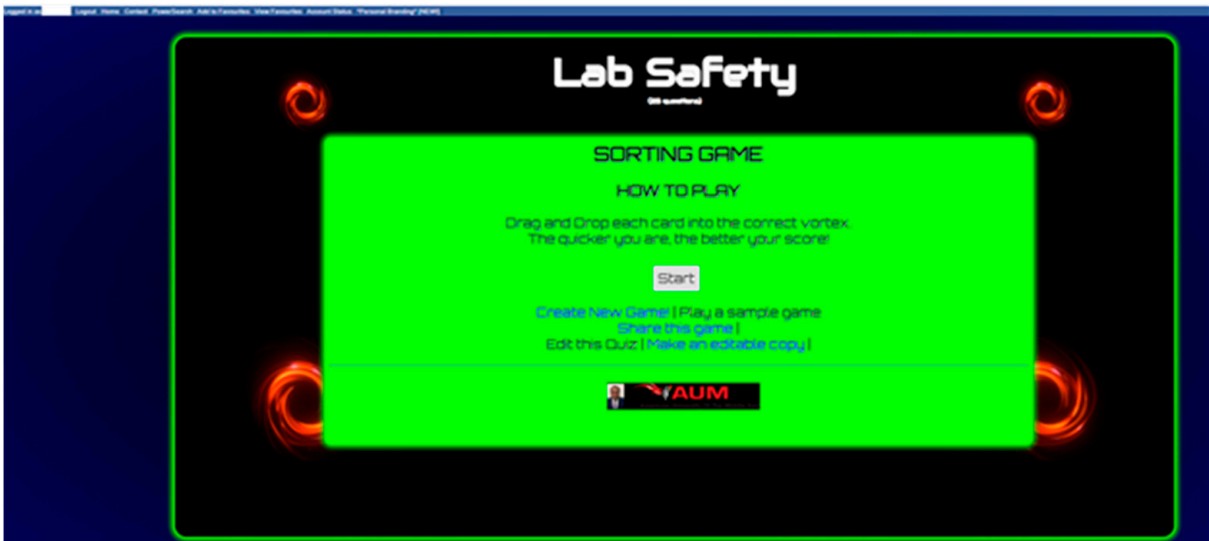

**Figure 1.** Screenshot of Vortex-based activity.

## 2. Materials and Methods

### 2.1. Research Subject

This study was conducted in two phases to achieve the objectives listed in the previous section. During the first phase, the research subject comprised over 276 students (aged 19–22 years) enrolled in five parallel classes of an undergraduate engineering seminar course taught by different instructors. This course was offered online during the spring semester of 2022 by the Department of Electrical Engineering, of which only 101 students participated in the survey, including both males (82.0%) and females (18.0%), and the majority of them belonged to the sophomore year. Similarly, more than 120 students registered for this course offered during the summer of 2022, and two parallel classes taught by the same teacher took part in the activities, of which 107 participated in the study, including both males (42.1%) and females (58.9%). It can be noted that a significantly higher number of students participated in the survey during the summer of 2022 because students take fewer courses during the relatively short summer semester and devote their time and effort to this course. In addition, this course does not have a final exam, meaning that students have more free time in the end and can afford to spend some time completing the survey. Furthermore, the instructor allocated time to complete the survey at the start and end of the lecture. The participants were under no obligation to participate in the research as part of their coursework, and it was completely voluntary. The objective was to explore the effect of game-based and gamified quizzes on student engagement and motivation. To promote acceptance and a judgment-free environment, student response systems were handled anonymously. Students had to complete the Vortex quiz activities using any name or ID number. The summer 2022 study was designed to examine gender differences in completing game-based tasks, and students were rewarded for completing activities, but no reward was offered for participating in the survey.

## 2.2. Study Design

This study is descriptive in nature and uses a survey research design to examine the status of transformation. The present study was conducted during the spring semester for sophomore students studying the "Electrical and Computer Engineering Sophomore Seminar" course toward a Bachelor of Electrical Engineering. This was borne out of the lecturer's (first author) desire to enhance the online lecture experience for engineering students with the overall goal of improving student engagement by providing active learning opportunities in the classroom. These activities can be referred to as competitive games as participants in competing games attempt to complete the task ahead of their opponents. These competition-based games are also called rousing games as they are used to energize students. Students can access these activities through the link provided on the course page on Moodle.

We based our study on the hypothesis that intrinsic motivation creates deeper engagement, which leads to deeper learning. Our focus was on motivating the students to engage with the learning material. Instrumental students pay more attention to the activities which can help them to improve their grades. In this context, students are less bothered to pay attention to such a course which does not help them in improving their GPA. It is a very challenging task to motivate the students to engage with the material in such a course, and this was the rationale behind the selection of this course for implementing game-based learning. This proposed weakness is actually the main strength of our study. If this instruction strategy proves successful in motivating and engaging students, then its success is guaranteed in a high-importance course.

## 2.3. Instruction Design

This study assesses the use of game-based and Moodle-based quiz activities (MBQAs) promoting active learning in the online setting of teaching. Game-based activities are created in accordance with Keller's [66] ARCS (attention, relevance, confidence, and satisfaction) model of motivation and self-determination theory (SDT) [67]. ARCS stresses that the teaching activities and materials must be matched with these four factors to improve the learning performance of the students and attain increased stimulation. Similarly, SDT is frequently used to comprehend human motivation in a variety of contexts, including games, employment, and education. According to SDT, people are most motivated when they feel in control of their conduct, when they are competent in what they do, and when they are linked to other people. Games that give players the chance to feel related, competent, and autonomous are more likely to be entertaining and increase player motivation. Individuals are less likely to be motivated when these requirements are not met, and they may also suffer from negative effects including stress, anxiety, and sadness. The GBQAs implemented in this study, shown in Table 1, are a relatively new experience for students which draws the attention of students' attention, piques their enthusiasm, and fosters a positive learning environment. These activities are related to the curriculum; they are played independently without intervention from the instructor, thus providing them with full control and resulting in confidence. Students feel satisfied when they answer a question correctly, finish an activity quickly, and see themselves performing better than their peers. The complexity of the topic, the timing, and the integration of an activity into a class were all taken into consideration when developing the activities. Each activity can be completed within one to three minutes, allowing it to incorporate more diverse activities without repetition and getting students bored.

A student's motivation is what propels them to act. The visible activity or proof of that motivation is engagement. Engagement requires motivation, yet successful engagement may also encourage students to remain motivated in the future. Meaningful engagement fosters deeper learning. These activities supplement the current instruction design, which is aligned with the course learning outcomes. Thus, completing these activities will help students in achieving the learning outcomes. In addition to being a goal in and of itself, student engagement in learning also serves as a vehicle for students to achieve successful

academic outcomes [68]. We can also measure how helpful these activities were in achieving the learning outcome. We had data in the form of assessment grades as this course has six assessments covering one or two. These assessment grades could be an indication if the student achieved learning outcomes or not, but the literature does not recommend using assessment grades to assess the student's learning as a result of these activities. This issue can be addressed in future work.

**Table 1.** List of activities.

| SR. # | Topic | Questions |
|---|---|---|
| A1 | Plan of study | 10 |
| A2 | Lab safety | 25 |
| A3 | Cover letter writing | 15 |
| A4 | Resume writing | 12 |
| A5 | Professional ethics | 21 |
| A6 | NSPE code | 12 |
| A7 | EDP | 20 |
| A8 | Making PowerPoint slides | 17 |
| A9 | Learning strategies | 12 |
| A10 | Writing report | 20 |

# means Serial Number in this Table.

### 2.4. Instrument

A short questionnaire comprising three parts was designed as a research instrument to collect data. The first part was designed to obtain demographic information from the participants. The second part used a five-point scale, ranging from strongly disagree (SD) to strongly agree (SA), to capture the opinions of the participants on 13 statements given in Table 2. The third part comprised open-ended questions to obtain the participants' unrestrained opinions. The first half of the questionnaire was focused on assessing engagement and motivation, and the second half was focused on the leaderboard. Q1 assessed the enjoyment imparted by the game. Q2, Q3, Q4, Q7, Q10, and Q11 assessed the perceived feeling of motivation, while Q3 and Q6 were a self-assessment of student learning. Q7–Q11 focused on assessing the impact of the leaderboard. Q12 and Q13 assessed students' engagement in and out of class. The reliability of the questionnaire calculated using Cronbach's alpha (CA) was found to be 0.835; thus, it was highly reliable.

Since this study explores the effectiveness of instruction design based on digital technology for engaging students, we considered both the technology acceptance model (TAM) [69] and the student engagement in schools' questionnaires [70] to gauge the degree to which students accepted the GBL and became engaged. The success of integrating game-based learning activities into the current teaching technique cannot be assured unless students accept it. According to the technology acceptance model (TAM), if students are prepared for the technology and it satisfies their expectations, it will be accepted. The simplicity of use and utility of technology are perceived using this paradigm, which might affect people's behavior toward embracing the technology. Candidate items 1, 8, 12, and 13 in the questionnaire indicated ease of use, while others were related to perceived usefulness. Similarly, SESQ is an instrument to measure engagement from a student perspective that is an outcome of the collaboration of researchers from 19 different countries. For the objectives of this study, only the items expressing engagement markers, such as affective (Q1, Q2, Q4, Q8, Q9) and behavioral (Q10–Q13) are considered.

**Table 2.** Survey questions.

| Number | Description |
| --- | --- |
| Question 1 | I had fun when playing the game. |
| Question 2 | The game-based activities motivated me to complete the class activities. |
| Question 3 | I believe that the games improved my understanding of the covered topics. |
| Question 4 | The game-based activities motivated me to arrive at class on time. |
| Question 5 | The game-based activities helped me to engage with the class material. |
| Question 6 | The game-based activities were beneficial to my overall learning. |
| Question 7 | I was more motivated to study the course material every week to do well on the leaderboard for the game-based activities. |
| Question 8 | I saw my comparison with the class. |
| Question 9 | It is important for me to see my position among my class fellows. |
| Question 10 | My comparison with the class fellows motivated me to study more. |
| Question 11 | Seeing my class fellow doing well made me study more. |
| Question 12 | I prefer to play game live during class time. |
| Question 13 | I also played this game outside lecture time just for fun. |

There was only one open-ended question for knowing the opinion of the participant:

- Mention anything you liked or did not like about game-based activity or if you have any suggestions.

We omitted negatively phrased questions to keep the questionnaire short, as, on the basis of our experience, lengthy questionnaires may demotivate students and they might not keep the complete questionnaire, thus affecting the completion rate in a negative way. There is a long tradition of including items in questionnaires that are phrased both positively and negatively to minimize extreme response bias and acquiescent bias. Despite published concerns about acquiescence bias, there is little evidence that the common practice of including both positive and negatively worded items solves the problem. The research conducted by Sauro and Lewis [71] found little evidence for these biases and reported that response bias effects are at best small and outweighed by the real effects of miscoding and misinterpreting by users. Furthermore, we used a five-point scale, ranging from strongly disagree (SD) to strongly agree (SA), to capture the opinions of the participants on 13 statements given in Table 2. Therefore, the negative statements are reflected by the scale.

*2.5. Research Procedure*

In the first experiment investigating the students' engagement, interactivity, and the impact of the leaderboard, all participants were divided into five different groups with 30–50 students in each group as shown in Table 3. The groups either completed the Vortex game-based activities or the MBQAs which were not designed using games. Furthermore, we made use of a leaderboard for some groups. During the second phase of the experiment, both groups completed the game-based activities only as we wanted to investigate the gender differences in the game-based instructions.

An online Survey Monkey form was used for data collection, and the link was shared with participants through the Moodle page and Moodle messages. The participants' responses were recorded. The questionnaires provided to the participants were anonymous, ensuring that they did not carry any information that could help identify them. Analyses of the data, frequency, percentage, mean, standard deviation, and survey results are discussed in Section 3. A thematic analysis was used to analyze the open-ended questions, and three main themes were identified using an inductive approach. Similar conceptual categories were merged to create larger overarching themes relevant to teachers' perceptions.

**Table 3.** Experiment and control groups.

| Control Group # | Semester | Mode | Leaderboard | Enrolled | Surveyed |
|---|---|---|---|---|---|
| TCG1 | Spring 2022 | GBQA/MBQA | No | 51 | 24 |
| TCG2 | Spring 2022 | GBQA | Yes | 30 | 29 |
| TCG3 | Spring 2022 | GBQA | No | 32 | 13 |
| TCG4 | Spring 2022 | MBQA | No | 49 | 17 |
| TCG5 | Spring 2022 | MBQA | Yes | 38 | 18 |
| TCG6 | Summer 2022 | GBQA | No | 66 | 63 |
| TCG7 | Summer 2022 | GBQA | No | 60 | 44 |

# means Control Group Number in this Table.

## 2.6. Research Ethics

The study was conducted by the Declaration of Helsinki and the protocol was approved by the Research Committee (RSV-004-03/08/2022). The survey was conducted online to ensure the anonymity of the participants. In addition, the participants were given an anonymous questionnaire with no data that could be used to identify the respondents.

## 3. Results

The collected data were analyzed using the R software version 4.0.3 [72]. First, we performed a univariate analysis including the descriptive statistics of the survey results. We then performed a bivariate analysis including the hypothesis tests for comparing the results of different groups in the activities. Regarding the range of studies attained, it can be seen that the vast majority of the participants belonged to the freshman year (47.5%), followed by the junior year (35.6%), and 6.9% belonged to the senior year. A breakdown of the participants is provided in Table A1 in Appendix A.

### 3.1. Activities Results

The activity completion rate for five different groups is shown in Figure 2. It can be seen that the completion rate of TCG1 was approximately 70% or higher for activities 2–5 which are game-based and dropped below 50% for subsequent activities which were Moodle-based. TCG2, assigned to game-based activities incorporating a leaderboard, initially had a completion rate of more than 60%, which dropped to as low as 30% later on. TCG3, game-based activities without a leaderboard, had an activity completion rate of approximately 70% or higher for the first six activities and then dropped below 40% for subsequent activities. Groups TCG4 and TCG5 (both groups assigned MBQA) had consistently low activity completion rates of approximately 20% or below. However, TCG5 had a higher activity completion rate than TCG4 did.

The average number of times that students completed activities is presented in Figure 3. It can be observed that the game-based groups had a higher access rate compared to the Moodle-based groups, indicating greater online engagement among students in the former. Furthermore, the access rate for TCG1 dropped to below 1.5 after switching from GBQAs to MBQAs after five activities. Additionally, TCG2 has a higher access rate compared to TCG3, suggesting that students in groups with a leaderboard tended to complete activities more frequently than those without a leaderboard.

Figure 4 presents the average scores of three groups, TCG1, TCG2, and TCG3, who completed game-based activities. The score recorded was the time taken to complete the activity, which serves as an indicator of performance, with lower scores indicating better performance. It can be observed that the group with the leaderboard (TCG2) outperformed the groups without a leaderboard, namely, TCG1 and TCG3, in nearly all activities, suggesting that the presence of a leaderboard may have positively impacted performance in these activities.

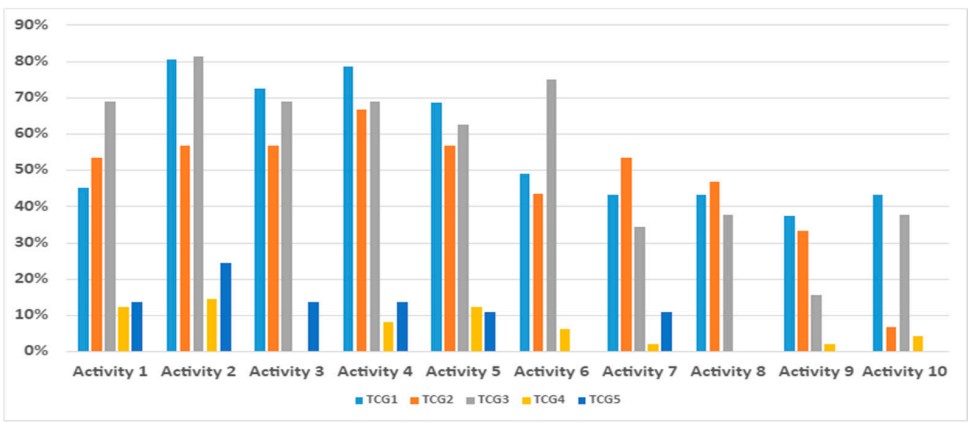

**Figure 2.** The activity completion rate for different groups during spring 2022.

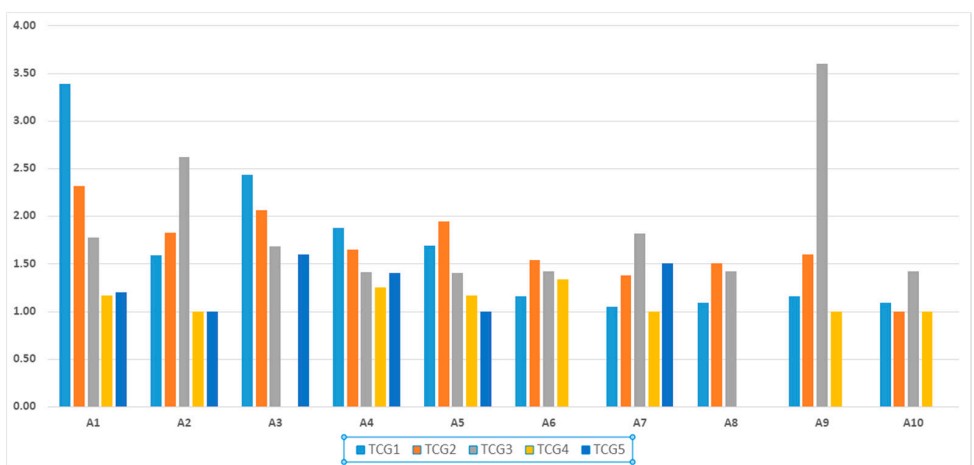

**Figure 3.** Number of times an activity was accessed by a student on average.

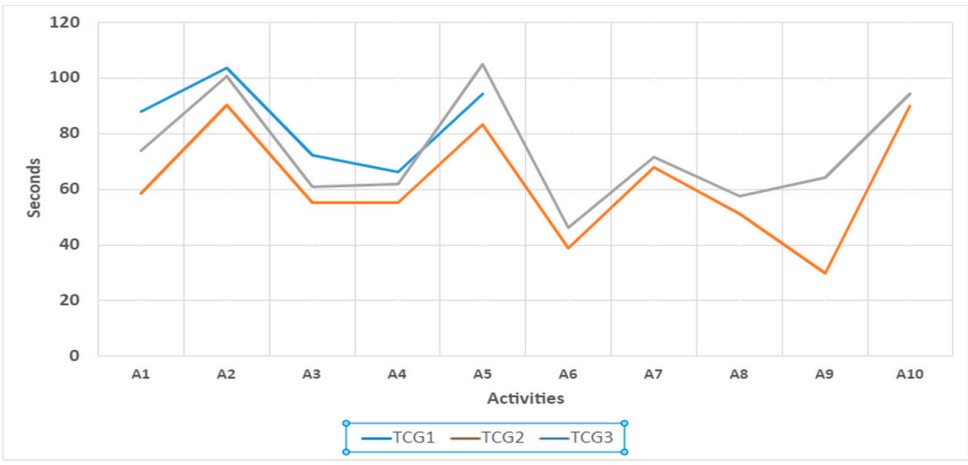

**Figure 4.** GBQA completion time comparison for different groups.

TCG2 completed the activities using Vortex with a leaderboard, while TCG3 (M3) completed the game-based activities without the leaderboard. It should be noted that a lower score for game-based activities is an indication of better performance. TCG3 performed slightly better than TCG2. We performed *t*-tests ($\alpha = 0.05$), and all *p*-values were greater than 0.05, indicating no statistically significant difference between the average activity scores for the groups with leaderboard and without leaderboard. The leaderboard

created a competition among the students, and the parameter measuring the performance was the time taken to complete the activities. Competition motivates the students to do well and reduces the time to complete the activities. This is only possible if students grasp the concepts. Thus, a lower score means that the student was motivated to do better.

We also collected data for Moodle-based online quiz activities. Table 4 summarizes the average scores and standard deviations in TCG5 (with leaderboard) and TCG4 (without leaderboard). It should be noted that a higher score for MBQAs indicates higher performance. We can see that TCG5 performed better than TCG4 in three out of four cases, showing that the leaderboard had an impact on student performance. We can observe that all *p*-values were greater than 0.05, indicating that there was no statistically significant difference between the average activity scores of TCG5 and TCG4. Thus, using a leaderboard in Moodle to complete the various game activities had no particular impact.

**Table 4.** Comparison of scores between groups that completed MBQA with and without a leaderboard.

| Activity | TCG5 | TCG4 | *p*-Value |
|---|---|---|---|
| A1 | 88.0 (8.37) | 65.0 (30.2) | 0.125 |
| A2 | 92.8 (10.9) | 83.6 (23.7) | 0.368 |
| A3 | 76.7 (22.4) | 91.7 (9.62) | 0.228 |
| A5 | 75.0 (16.7) | 63.9 (26.7) | 0.441 |

The results provided in Table 5 represent a comparison of the activity completion times of the male and female groups. Activity completion times were higher for females than for males for every game activity, except for the first activity. Completing the activity related to lab safety was the most time-consuming for both genders. On the other hand, the activity related to making PowerPoint slides was completed in half the time required for lab safety activities. The *p*-values indicated a statistically significant difference between the average score for the activity regarding cover letter writing ($p$-value = 0.001), resume writing ($p$-value < 0.001), professional ethics ($p$-value = 0.004), and the engineering design process ($p$-value = 0.001) in the two groups. These results suggest that male users could complete these activities significantly faster than female users.

**Table 5.** Activity completion times for female and male participants.

| Activity | TCG7 (Male) | | | | TCG6 (Female) | | | |
|---|---|---|---|---|---|---|---|---|
| | **Mean** | **SD** | **Min.** | **Max.** | **Mean** | **SD** | **Min.** | **Max.** |
| A1 | 73.04 | 52.50 | 17 | 232 | 73.16 | 73.50 | 21 | 351 |
| A2 | 103.72 | 46.76 | 45 | 204 | 110.87 | 59.38 | 45 | 321 |
| A3 | 68.74 | 34.35 | 25 | 175 | 97.36 | 51.92 | 30 | 336 |
| A4 | 48.04 | 23.55 | 22 | 141 | 84.27 | 57.96 | 33 | 438 |
| A5 | 83.20 | 25.80 | 36 | 166 | 102.95 | 41.28 | 52 | 253 |
| A7 | 78.90 | 21.75 | 46 | 138 | 105.14 | 47.01 | 42 | 314 |
| A8 | 51.67 | 25.59 | 27 | 161 | 56.59 | 23.84 | 31 | 152 |

*3.2. Survey Results*

The distribution of the survey results is shown in Figures 5 and 6 for spring 2022 and summer 2022, respectively. A detailed breakdown of responses based on demographics is provided for spring 2022 and summer 2022 in Table A1 in Appendix A.

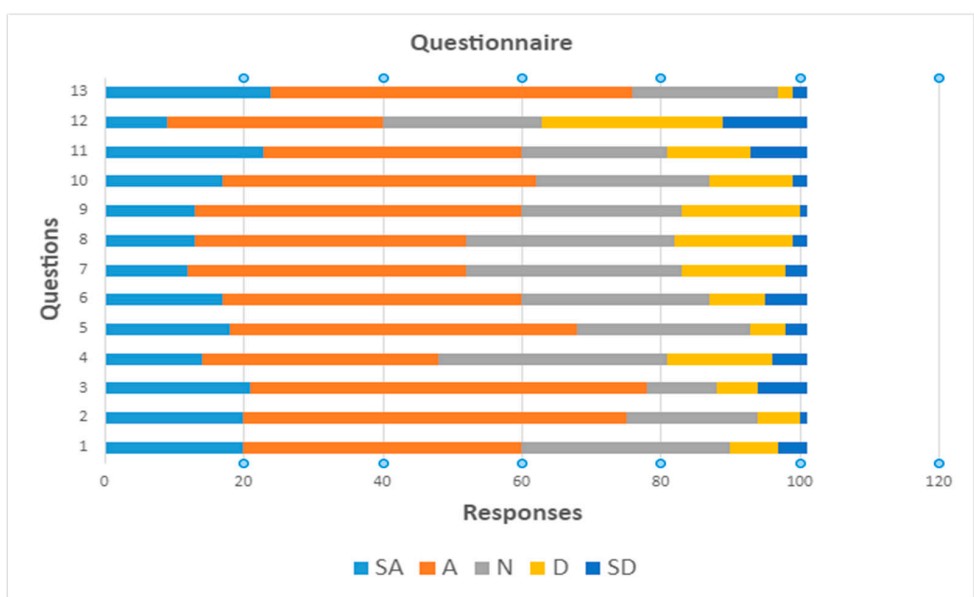

**Figure 5.** Responses to survey questions during spring 2022.

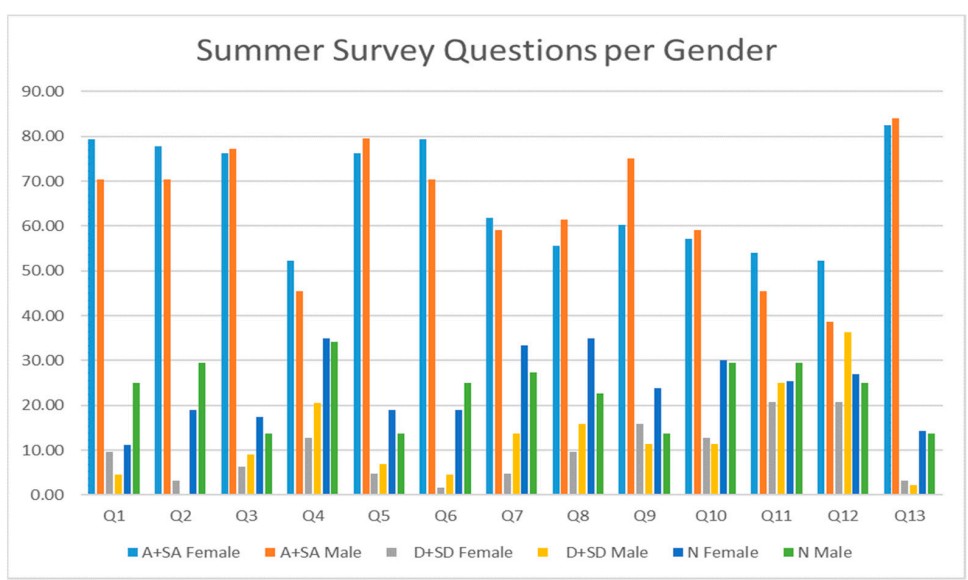

**Figure 6.** Comparison of questionnaire responses from different genders from summer 2022.

In response to question 1, only 59.94% of the participants (mean = 2.92; SD = 1.24) reported having fun while playing the game to complete the GBQAs, and 10.89% did not find it enjoyable at all. Interestingly, a higher percentage of females (83.3%) compared to males (54.2%) found it fun. In the summer, 58.59% of participants agreed that the game was enjoyable, with 74.9% of females responding positively compared to 79.4% of males. It is noteworthy that around 40% of the participants did not agree that they had fun, with 30% expressing a neutral stance. This finding is surprising, as games are generally believed to be enjoyable and memorable for most people, making them effective tools for learning. It can be speculated that students with high expertise levels may not have found the game challenging, which could have influenced their perception of fun. When asked about their motivation to complete the activities, the responses showed that 74.25% of participants in spring (mean = 2.92; SD = 1.22) and 74.76% in summer felt motivated. Interestingly, a higher percentage of females reported feeling motivated compared to male students, both in spring (88.9% vs. 71.1%) and in summer (77.8% vs. 70.5%).

Around 77% of the participants for both phases thought that these activities improved their understanding of the covered topic and, again, more females agreed to this point as compared to male counterparts for spring, 88.9%, and 74.7%, respectively. The ratio was the same for both genders in the summer. A very low number of participants, around 18% during spring and 49.5% during summer, agreed when asked if the activities motivated them to arrive in class on time. An equal percentage of both males and females had this opinion during spring and summer. The results related to the engagement with class material showed that only 67.73% of participants during spring and 77.5% during summer agreed that the game-based activities helped them to engage with the class material, while 24.75% disagreed, and the others were neutral. More males agreed than females in spring (68.7% vs. 79.5%) and summer (79.5% vs. 76.2%). For spring data, around 60% of participants thought that these activities were beneficial to their overall learning, while 13.86% disagreed and others were neutral. A slightly higher percentage of males agreed than females, 60.2% and 55.6%, respectively. For summer, around 75% of the participants thought that these activities were beneficial to their overall learning, while 21.49% were neutral. The percentage of males who agreed was lower than that of females, 70.5% and 79.4%, respectively. Only half (52%) of the participants during spring felt motivated to study the course material every week to do well on the leaderboard, and this number was 60.74% during summer. Female students were more determined to do well than males, with agreement rates of 72.2% and 47%, respectively. The gap narrowed during the summer, and the response was around 62% and 59% during the summer. More senior students agreed (71.4%) than the sophomore (50.9%) and junior (50%) students. Only slightly more than half (51.28% during spring and 57.94% during summer) of the participants confirmed that they saw their comparison with the class, and 30% were neutral for both spring and summer. A higher female proportion (61.1%) saw their comparison than males (49.4%) in spring, and this trend reversed in summer (55.6% females and 61.4% males).

Around ~60% during spring and 66.35% during summer agreed that it was important for them to see their position among the class, whereas around 18% during spring and 14% during summer disagreed, and the remainder were neutral. A higher number of sophomore students (66.7%) agreed than others. More than 60% of participants registered in spring agreed that their comparison with the class fellows motivated them to study more, while 14% disagreed, and others were neutral. The response was similar in the summer, where 57.94% agreed and 29.91% were neutral. The gender response was the same both in spring and summer. In response to the question of whether seeing their class fellows made them study more, around 60% of participants agreed, 20% disagreed, and others were neutral during spring. Overall, 50% of females and 61.4% of males responded positively. For the summer survey, 50.46% agreed and 22.42% disagreed. The percentage of female (54%) participants agreeing was more than males (45.5%).

Students did not like to play the game during lecture time as only 39.60% and 46.72% of participants said so for spring and summer surveys, respectively. More females said so during spring (50.4% females vs. 37.3% males) and during summer (52.4% females vs. 38.6% males), and this proportion was higher for females and sophomore students. The majority of the participants who took part in the survey during spring 2022 (around 75%) played this game outside the lecture hour, with 22% being neutral. There was no major discrepancy among the responses of participants belonging to different demographics.

There were two survey questions where the participants had the opportunity to state their preferences regarding the GBQAs or MBQAs.

Q14: Which of the activities did you enjoy more?

Q15: Which one would you recommend for future use?

More than half of the participants enjoyed game-based activities (68.32%), and they recommended it for future use (67.33%).

*3.3. Analysis of Open-Ended Survey Responses*

The opinions of the survey participants regarding game-based activities, including their suggestions, were assessed through open-ended questions. Open-ended questions do not restrict the participants to a set of predetermined answer options; therefore, they allow for gathering a wide range of sincere opinions from the survey subject, but they demand more time, consideration, and thought from the respondent. Therefore, some participants abstained from responding or responded limitedly, making the responses hard to analyze. The responses were imported to R, and thematic analyses, including text-mining processes, were carried out. Similar responses based on commonly used frequent keywords were grouped, forming three main categories.

Theme 1: Game-based activities are a good way to review the course material.

Thematic analysis revealed that the students found game-based activities useful because they helped them to provide an overview of the topics studied during the lectures.

"What I like is that it is a new way to understand and review the course material."

"What I liked about the game-based activity is that it is a fantastic method to review lecture material."

Theme 2: Game-based activities help students to become more engaged in the lectures.

The students felt more engaged when participating in game-based activities which helped them focus their attention on the lecture topics. This was also evident from the student responses given below.

"Great idea to engage the students in the class material."

"It is good to use live games where all students get engaged at the same time with the same question and to see who solves them faster and correctly."

Theme 3: Playing games was a fun and satisfying experience.

The thematic analysis revealed that playing the game-based activities created a positive atmosphere in the classroom, which could help students to perform better.

"I enjoyed playing the games; they helped to build a better connection between me and my class and professor. It was fun interacting with everyone when playing the game."

"Winning the game was so satisfying."

## 4. Discussion

RQ1: How are students' engagement and motivation affected by the usage of game-based activities?

The analysis of activity data and survey responses revealed that GBQAs were more effective in engaging students in the educational process compared to MBQAs. This is evident from the higher activity completion rate (Figure 2), the average activity access rate (Figure 3), and the responses to the survey questions. It is also clear from Table A1 that groups completing game-based elements said that they were more engaged and had slightly more fun than those completing Moodle-based games. The higher completion rate for GBQAs compared to Moodle-based learning suggests that game-based learning was more engaging for students. Additionally, the higher average completion rate indicates that students were more actively involved in the game, which aligns with the findings of previous studies (e.g., [20,29,62]). Furthermore, students accessed game-based activities more frequently than MBQAs, which could be for entertainment purposes or to improve their scores for self-satisfaction. This highlights another important aspect of game-based learning, as students who initially struggled may be motivated to improve and develop a growth mindset, as discussed in Dostal's study (2015) [73]. Consequently, students with a growth mindset are more likely to attempt new approaches multiple times until they succeed.

During the spring semester, over 60% of participants and, during the summer semester, 81% of participants expressed that they had fun while completing game-based activities, which can serve as an indicator of engagement. Fun is commonly recognized as a crucial element of in-game engagement, as it signifies that players are enjoying the gameplay and are emotionally invested in the experience. When players have fun, they are more

likely to continue playing, explore the game world, and be motivated to achieve in-game goals [32]. The responses from Q5 demonstrate that a significant number of participants found the game-based activities engaging. This innovative approach of using gaming elements to present the material and challenge the students was successful in attracting their attention and engaging them with the learning material [33]. Similarly, a large number of participants during both phases were of the view that the games improved their understanding of the covered topics, revealing that students were engaged in meaningful ways through these game-based activities. When individuals have a clear understanding of a topic or concept, they are more likely to engage with it. Understanding provides a foundation for meaningful engagement, as it enables individuals to connect with the subject matter, ask questions, and participate in discussions or activities related to it. Students who understand the material being taught are more likely to actively participate in discussions, ask questions, and complete assignments, leading to higher levels of engagement. From the reciprocal perspective, meaningful engagement can deepen and reinforce understanding. When individuals are actively engaged in an activity, they are more likely to process and internalize the information, leading to a better understanding of the subject matter.

Analysis of Figure 2 revealed that the activity rate for groups completing GBQAs was much higher than that for MBQAs. This indicates that game-based activities are an effective strategy to motivate and engage students in learning, which is in line with the findings of [34,55,57,60]. This is also evident from the statistics of TCG1, which had a higher completion rate for the first half of GBQAs and a lower completion rate for MBQAs. The use of GBQAs introduces variety and novelty into the learning experience, which can help prevent monotony and increase students' motivation. The interactive and dynamic nature of the platform created a fresh and engaging learning environment, which may pique students' curiosity and motivation to explore and learn. It should be noted that this is a zero-credit hour course with a pass or fail as a possible outcome. This means that students had no extrinsic motivation to complete the activities; hence, we can safely claim that they were intrinsically motivated to complete these activities. It can be seen that the activity completion rate dropped in the latter half, which can be attributed to several reasons. First, some students might have become bored after completing a few activities because of simple game mechanics and considering the founding elements of educational video games include fantasy, curiosity, challenges, and power when designing games [74]. To sustain students' interest, it is important to introduce new information so that students feel challenged. However, designing such activities might overload the teacher, and the nature of courses does not allow for the luxury of spending huge time on gaming due to its disruptive nature. In addition, some teachers have no prior gaming experience, which makes it difficult for them to modify it for educational reasons [75]. Second, as these activities did not carry any significant weight toward the final mark, some students may have viewed them as unnecessary. Thirdly, the majority of students knew about the possible outcome of their results and did not pay much attention to this course as it would not have any impact on their grades, and they concentrated their efforts on other courses whose outcomes could improve their grade point average (GPA). It can also be noted that TCG5 had higher completion rates than TCG4, meaning that the leaderboard did motivate slightly. Furthermore, as mentioned earlier, students were engaged more through GBQAs, and such meaningful engagement promotes a sense of ownership and responsibility for the learning process. When individuals are actively involved, they are more motivated to take ownership of their learning and are more likely to be invested in the outcomes. This can result in increased intrinsic motivation and a deeper sense of commitment to the subject matter. This is also evident from the student response to question 13, where an overwhelming majority of the students declared that they played the games outside the lecture hour, which is an indication of intrinsic motivation. This also indicates that the activities created interest in the subject matter and students were willing to go beyond the required class hours to do well. Students can typically choose their own pace, select topics or levels of difficulty, and have the flexibility to revisit and

review content as needed. This autonomy can foster a sense of ownership and control over their learning, which can positively impact their motivation. Another aspect of GBQAs is the immediate feedback, which allows students to quickly assess their progress and make improvements. This instant feedback can boost students' motivation by providing them with a clear understanding of their strengths and weaknesses and encouraging them to strive for better performance.

RQ2: What is the impact of the leaderboard on student motivation and behavior?

The analysis of the average completion time for the game-based activities for three different groups provided in Figure 4 reveals that the group with an additional gamification element as a leaderboard performed better, which may have resulted in higher task performance than those without a leaderboard. A similar trend can be observed from the data provided in Table 4, where TCG2 performed slightly better than TCG3. Furthermore, it can be seen in Table A1 that groups TCG2 and TCG5 has more fun playing and were more motivated than the groups not using the leaderboard. This means that the leaderboard can motivate individuals or teams to perform better and strive for higher rankings [53,54]. The visual representation of rankings and scores on a leaderboard can trigger a desire to improve one's position and achieve recognition, leading to increased effort and engagement with the task. The desire to be at the top of the leaderboard can create healthy competition and drive individuals to put in more effort. Knowing that their performance is being tracked and compared to others can encourage individuals to stay on track and perform to the best of their abilities. In some cases, leaderboards can foster a sense of achievement, provide a clear goal to strive for and create a positive feedback loop that encourages learners to perform better. Furthermore, the time to complete activities was higher for groups without leaderboards (i.e., TCG1 and TCG2), indicating that these students were not motivated to do well. The use of social features such as a leaderboard promotes social interaction and competition among students. This social element can create a sense of community, foster friendly competition, and increase motivation through social recognition and support.

However, the impact of the leaderboard is not the same for everyone and depends on multiple factors, including the specific context, design, and implementation of the leaderboard system, as well as individual learner characteristics. This is evident from the responses to Q7–Q11 as many students did not have a favorable opinion toward the use of the leaderboard. For example, in response to question 11, both groups with leaderboards, TCG2 and TCG5, did not agree that seeing others doing well motivated them. For some students, leaderboards may create a sense of pressure or anxiety, leading to a negative impact on their performance. It can also result in students focusing solely on achieving a high rank on the leaderboard, rather than engaging deeply with the learning content or developing a deep understanding of the material.

While leaderboards can be motivating for some individuals, they can also be demotivating for others. Individuals who are consistently ranked lower may become discouraged and give up, even if they are making progress and improving in other ways. Leaderboard can create a culture of winners and losers, which may not be conducive to a positive work environment. Students are less inclined to share knowledge or help each other, as they may perceive others as competitors on the leaderboard. This can negatively impact peer-to-peer learning, collaboration, and cooperation. This can be observed by the fact that a very small number of the students responded positively to questions Q10 and Q11, which were related to the leaderboard. It is also evident from the activity completion rate provided in Figure 2 that group TCG3 without a leaderboard had a higher activity completion rate compared to the group with the leaderboard. In this research, we used absolute/infinite leaderboards where all users and their scores are displayed on the Moodle page [19], giving players at the top a greater feeling of accomplishment than players at the bottom. This has a negative impact as it is not inclusive and poses a threat to introverted students as they feel that they are forced into the competition, which may impact their attitude negatively. Students who believe they will never reach the top are more likely to give up on their goals and

might not take part in them at all, thus missing an opportunity to learn [76]. Students who do not perform well or who are consistently at the bottom of the leaderboard may experience decreased self-esteem or feelings of inadequacy, which can negatively impact their motivation to continue participating or learning.

RQ3: Do males or females have the same perception toward game-based activities?

In terms of motivation for completing game-based activities, 77.8% of the female gender answered affirmatively, compared to 70.5% of the male gender. In other words, a greater number of females were motivated than males. This finding corresponds with that of [77] implying that female players are considerably more motivated than male players in a digital game of a moderate genre. The results from the comparison of activity completion time for both genders, presented in Table 5 indicate that there was a difference in gameplay behavior between males and females. On average, male students were slightly faster in completing the activity compared to their female counterparts. This finding aligns with a study conducted by [78], which reported that female students consistently spent more time than male students on gameplay across different studies. Additionally, research by [79] suggested that female students tend to prioritize goal clarity and social interaction as more important factors in digital learning games, while male students tend to place more emphasis on the challenge, progress feedback, and visual appeal in such games. These findings highlight gender differences in gameplay behavior and preferences among students, which could have implications for designing effective digital learning games that cater to the diverse needs and preferences of learners of different genders.

When we analyzed the survey results of summer 2022, which contained a more balanced proportion of students of both genders than those of spring 2022, it was revealed that females enjoyed playing games more than males, and the same was evident from the spring 2022 outcomes. Although more males (58.5%) play video games than females (41.5%), as per the distribution of video game users in the United States in 2021, the enjoyment of digital games is ultimately a personal preference and cannot be generalized by gender. Males favor gameplay that involves strategy and clear directions from the system, in contrast to females, who favor games that encourage exploration. Moreover, males view digital games as a medium for socializing and skill development, whereas women prefer to adopt autonomous learning. Unfortunately, Vortex-based quizzes lack strategies but do have socialization features in the form of scores and leaderboards. All of the groups had explicit leaderboards, due to which male students had slightly less fun and motivation while playing games. This discovery concurs with that of Joiner et al. [80], whose findings showed that gender differences in digital game preferences have an impact on each group's degree of motivation.

The responses to Q9 indicate that male students were more inclined to the comparison with other students than females. In other words, male students were more inclined toward the social comparison in this particular context in order to assess their competence. The competitive nature of the leaderboard might have urged them to make more upward comparisons and try to improve themselves compared to females who liked to have more downward comparison. On the basis of the responses to Q12 of the summer 2022 survey, it appears that a higher number of males compared to females did not prefer to complete activities during class time. This could be attributed to various factors, such as gender stereotypes, perceived lack of challenge, technological limitations, compulsory nature of the activity, or personal learning preferences. Some male students may feel societal pressure to conform to traditional gender norms and stereotypes that view game-based activities as childish or not serious. They may worry about being perceived as immature or not conforming to traditional masculine expectations, thus choosing to avoid such activities to prevent potential social judgment. Additionally, some males may not have found the games challenging enough or may have faced limitations in accessing a stable internet connection, which could have impacted their engagement. Moreover, the game-based activities may not have aligned with the preferences or learning needs of some male students. Furthermore, if the game-based activities were conducted during class time, some students may have

viewed them as compulsory without any perceived benefit or reward, which could have resulted in a lack of motivation to participate for instrumental students. However, some female students may have viewed in-class activities as an opportunity to demonstrate their abilities and engage in healthy competition with male students, potentially influencing their motivation to participate.

*Limitations of Study*

This study had several potential limitations. Firstly, all groups were not taught by the same teacher, which may have introduced inconsistency in the delivery of instruction and impacted the continuity and coherence of the study. This limitation was encountered in the subsequent phase where different groups were taught by different teachers. Additionally, some students considered playing games during class time as an alternative approach to roll call during spring 2022, despite being explained that it was optional activity and did not count for anything in terms of attendance and grades. These misconceptions may have influenced their engagement and opinion in the study. Moreover, the Vortex sorting game used in the study was chosen for its simplicity, but it may not have been challenging enough for engineering students, which could have affected their motivation and engagement. Lastly, the study was conducted in a single course at one university with a similar cultural background, and the findings may have limited generalizability. Conducting the study with participants from diverse cultures and backgrounds could enhance the meaningfulness of the results.

## 5. Conclusions

We utilized game-based quizzes implemented using the Vortex sorting game to assess the engagement and motivation of undergraduate students enrolled in a sophomore seminar course. The findings indicate that incorporating digital games into engineering education can effectively motivate students and sustain their interest in the course material. Analysis of anonymous feedback surveys revealed positive responses from students toward the game-based activities. Although not statistically significant, the gamified Vortex activities were found to have a more positive impact on student engagement and motivation. However, the use of a leaderboard as a gaming element was observed to have both positive and negative effects, improving the performance of some students while demotivating others. The favorable response from students toward game-based activities suggests that integrating games into undergraduate general engineering lectures can be an enjoyable and engaging experience. We also observed some differences in perception of game-based activities between genders, with female students showing slightly higher levels of engagement and motivation compared to their male counterparts, but not preferring to compare themselves with other students. In summary, the instructional approach and design of game-based activities need careful consideration and support from academia, with a focus on creating games that offer appropriate challenge levels, provide clear instructions, relevant content, and meaningful feedback, and align with the preferences and learning needs of male students in particular. Additional training and resources may be needed to ensure the effective implementation of game-based activities in engineering education.

**Author Contributions:** Conceptualization, M.N.; methodology, M.N. and W.F.; software, M.O.; validation, M.N. and W.F.; formal analysis, M.N., M.O. and W.F.; investigation, M.N. and M.O.; resources, M.N.; data curation, M.N.; writing—original draft, M.N. and M.O.; writing—review & editing, W.F.; visualization, M.N.; supervision, M.N.; project administration, M.N.; All authors have read and agreed to the published version of the manuscript.

**Funding:** This research received no external funding.

**Data Availability Statement:** Not applicable.

**Conflicts of Interest:** The authors declare no conflict of interest.

# Appendix A

**Table A1.** Breakdown of responses for spring and summer 2022 surveys.

| Q.# | Resp. | Spring 2002 | | | | | | | Summer 2022 | |
|-----|-------|-------------|---|---|---|---|---|---|-------------|---|
| | | Female | Male | TCG1 | TCG2 | TCG3 | TCG4 | TCG5 | TCG6 | TCG7 |
| Q1 | SA + A | 15 (83.3%) | 45 (54.2%) | 14 (58.34%) | 19 (65.52%) | 8 (61.54%) | 9 (52.94%) | 11 (61.11%) | 50 (79.4%) | 31 (70.5%) |
| | D + SD | 2 (11.10%) | 9 (10.8%) | 5 (20.83%) | 8 (27.59%) | 4 (30.77%) | 6 (35.29%) | 5 (27.68%) | 6 (9.5%) | 2 (4.5%) |
| | N | 1 (5.6%) | 29 (34.9%) | 5 (20.83%) | 2 (6.89%) | 1 (7.69%) | 2 (11.77%) | 2 (11.11%) | 7 (11.1%) | 11 (25.0%) |
| Q2 | SA + A | 16 (88.9%) | 59 (71.1%) | 14 (58.33%) | 24 (82.76%) | 9 (69.23%) | 12 (70.59%) | 16 (88.89%) | 49 (77.8%) | 31 (70.5%) |
| | D + SD | 1 (5.56%) | 6 (7.23%) | 5 (20.83%) | 5 (17.24%) | 3 (23.08%) | 5 (29.41%) | 1 (5.56%) | 2 (3.2%) | 0 (0.0%) |
| | N | 1 (5.56%) | 18 (21.7%) | 5 (20.83%) | 0 (0.00%) | 1 (7.69%) | 0 (0.00%) | 1 (5.56%) | 12 (19.0%) | 13 (29.5%) |
| Q3 | SA + A | 16 (88.9%) | 62 (74.7%) | 20 (83.33%) | 23 (79.31%) | 9 (69.23%) | 12 (64.70%) | 14 (77.78%) | 48 (76.2%) | 34 (77.3%) |
| | D + SD | 1 (05.6%) | 12 (14.0%) | 0 (0.00%) | 3 (10.34%) | 3 (28.08%) | 2 (17.65%) | 2 (11.11%) | 4 (6.3%) | 4 (9.09%) |
| | N | 1 (05.6%) | 9 (10.8%) | 4 (16.67%) | 3 (10.34%) | 1 (7.69%) | 3 (17.65%) | 2 (11.11%) | 11 (17.5%) | 6 (13.6%) |
| Q4 | SA + A | 9 (50.0%) | 9 (47.00%) | 15 (62.50%) | 10 (34.48%) | 4 (30.77%) | 10 (58.82%) | 9 (50.00%) | 33 (52.4%) | 20 (45.5%) |
| | D + SD | 3 (16.7%) | 17 (20.5%) | 5 (20.83%) | 11 (37.93%) | 7 (53.85%) | 4 (23.53%) | 6 (33.33%) | 8 (12.7%) | 9 (20.4%) |
| | N | 6 (33.3%) | 27 (32.5%) | 4 (16.67%) | 8 (27.59%) | 2 (15.38%) | 3 (17.65%) | 3 (16.67%) | 22 (34.9%) | 15 (34.1%) |
| Q5 | SA + A | 11 (61.1%) | 57 (68.7%) | 18 (65.00%) | 21 (72.41%) | 10 (61.54%) | 10 (58.82%) | 11 (55.55%) | 48 (76.2%) | 35 (79.5%) |
| | D + SD | 1 (05.6%) | 7 (8.4%) | 4 (16.67%) | 6 (20.69%) | 3 (23.08%) | 4 (23.53%) | 6 (33.33%) | 3 (4.8%) | 3 (6.8%) |
| | N | 6 (33.30%) | 19 (22.9%) | 2 (8.33%) | 2 (6.90%) | 2 (15.38%) | 3 (17.65%) | 1 (5.56%) | 12 (19.0%) | 6 (13.6%) |
| Q6 | SA + A | 10 (55.6%) | 50 (60.2%) | 15 (62.50%) | 17 (58.62%) | 7 (53.85%) | 11 (64.71%) | 10 (55.56%) | 50 (79.4%) | 31 (70.5%) |
| | D + SD | 2 (11.1%) | 12 (14.50%) | 4 (16.67%) | 9 (31.03%) | 4 (30.77%) | 4 (23.53%) | 6 (33.33%) | 1 (1.59%) | 2 (4.55%) |
| | N | (33.3%) | 21 (25.3%) | 5 (20.83%) | 3 (10.34%) | 2 (15.38%) | 2 (11.76%) | 2 (11.11%) | 12 (19.0%) | 11 (25.0%) |
| Q7 | SA + A | 39 (47.0%) | 0 (0.0%) | 13 (54.17%) | 11 (37.93%) | 5 (38.46%) | 10 (58.82%) | 13 (72.22%) | 39 (61.9%) | 26 (59.1%) |
| | D + SD | 16 (19.3%) | 0 (00.0%) | 6 (25.00%) | 13 (44.83%) | 4 (30.77%) | 5 (29.41%) | 3 (16.67%) | 3 (4.76%) | 6 (13.6%) |
| | N | 3 (16.70%) | 28 (33.7%) | 5 (20.83%) | 5 (17.24%) | 4 (30.77%) | 2 (11.76%) | 2 (11.11%) | 21 (33.3%) | 12 (27.3%) |
| Q8 | SA + A | 11 (61.1%) | 41 (49.4%) | 15 (62.50%) | 10 (34.48%) | 6 (46.15%) | 10 (58.82%) | 11 (61.11%) | 35 (55.6%) | 27 (61.4%) |
| | D + SD | 15 (18.1%) | 0 (0.0%) | 5 (20.83%) | 11 (37.93%) | 5 (38.46%) | 6 (35.29%) | 3 (16.67%) | 6 (9.52%) | 7 (15.9%) |
| | N | 3 (16.7%) | 27 (32.5%) | 4 (16.67%) | 8 (27.59%) | 2 (15.38%) | 1 (5.88%) | 4 (22.22%) | 22 (34.9%) | 10 (22.7%) |
| Q9 | SA + A | 11 (61.1%) | 49 (59.0%) | 15 (62.50%) | 16 (55.17%) | 7 (53.85%) | 11 (64.71%) | 11 (61.11%) | 38 (60.3%) | 33 (75.0%) |
| | D + SD | 3 (16.7%) | 15 (18.1%) | 4 (16.67%) | 7 (24.14%) | 5 (38.46%) | 3 (17.65%) | 4 (22.22%) | 10 (15.9%) | 5 (11.4%) |
| | N | 4 (22.20%) | 19 (22.90%) | 5 (20.83%) | 6 (20.69%) | 1 (15.38%) | 3 (17.65%) | 3 (16.67%) | 15 (23.8%) | 6 (13.6%) |
| Q10 | SA + A | 11 (61.1%) | 51 (61.4%) | 14 (58.33%) | 18 (62.07%) | 7 (53.85%) | 12 (70.59%) | 11 (61.11%) | 36 (57.1%) | 26 (59.1%) |
| | D + SD | 1 (5.56%) | 13 (15.7%) | 4 (16.67%) | 8 (27.59%) | 4 (30.77%) | 3 (17.65%) | 6 (33.33%) | 8 (12.7%) | 5 (11.4%) |
| | N | 6 (33.3%) | 19 (22.9%) | 6 (25.00%) | 3 (10.34%) | 2 (15.38%) | 2 (11.76%) | 1 (5.56%) | 19 (30.2%) | 13 (29.5%) |
| Q11 | SA + A | 9 (50.0%) | 51 (61.4%) | 16 (66.67%) | 14 (48.28%) | 9 (69.23%) | 12 (70.59%) | 9 (50.00%) | 34 (54.0%) | 20 (45.5%) |
| | D + SD | 5 (27.8%) | 15 (18.1%) | 4 (16.67%) | 7 (24.14%) | 3 (23.08%) | 3 (17.65%) | 4 (22.22%) | 13 (20.6%) | 11 (25.0%) |
| | N | 4 (22.2%) | 17 (20.5%) | 4 (16.67%) | 8 (48.28%) | 1 (7.69%) | 2 (11.76%) | 5 (27.78%) | 16 (25.4%) | 13 (29.5%) |
| Q12 | SA + A | 9 (50.0%) | 31 (37.3%) | 10 (41.67%) | 9 (31.03%) | 2 (15.38%) | 10 (58.82%) | 9 (50.00%) | 33 (52.4%) | 17 (38.6%) |
| | D + SD | 4 (22.2%) | 34 (41.0%) | 5 (20.83%) | 6 (20.69%) | 5 (38.46%) | 2 (11.76%) | 5 (27.78%) | 13 (20.6%) | 16 (36.4%) |
| | N | 5 (27.8%) | 18 (21.7%) | 9 (37.50%) | 14 (48.28%) | 6 (46.15%) | 5 (29.41%) | 4 (22.22%) | 17 (27.0%) | 11 (25.0%) |
| Q13 | SA + A | 14 (77.8%) | 62 (74.7%) | 17 (70.83%) | 22 (75.86%) | 9 (69.23%) | 14 (82.35%) | 14 (77.78%) | 52 (82.5%) | 37 (84.1%) |
| | D + SD | 1 (5.56%) | 3 (03.6%) | 6 (25.00%) | 5 (17.24%) | 4 (30.77%) | 3 (17.65%) | 3 (16.67%) | 2 (3.2%) | 1 (2.3%) |
| | N | 3 (16.7%) | 18 (21.7%) | 1 (4.17%) | 2 (6.90%) | 0 (0.00%) | 0 (0.00%) | 1 (5.56%) | 9 (14.3%) | 6 (13.6%) |

# means the number of the question in the Table.

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
