# Peer review of "Effect of Digital Game-Based Learning on Student Engagement and Motivation"

_computers, doi:10.3390/computers12090177_

Round 1

Reviewer 1 Report

This is a very interesting and valuable paper in the field of game-based learning, gamification, and student interest/motivation/engagement improvement field.

I liked the paper very much and enjoyed reading it, although the results part was a bit demanding - a lot of numbers and percentages one following the other... the reader can get lost very easily.

I have just a few remarks which should be considered by the authors:

- the paper starts with a bit bold presumption: "The world of academia is currently facing a serious crisis, a crisis of engagement". While I personally and intimately agree, this statement should be supported by a citation (or, even better a few of them) since it is a basis for the whole research and this paper in the end. 

- the authors calculated Cronbach's alpha for their questionnaire reliability, but the questions seem to be rather diverse, asking different things (e.g. q8 looks like a true/false question, and q12 and q13 are more "location" based), are you sure you can calculate Cronbach's alpha for the whole questionnaire?

- the authors discuss the "likeness" of the game (Vortex sorting game), which is rather simple and could be considered "too simple" or "boring" by some (or many?) of the students. Maybe some of the results are influenced by the nature of the game?

- the results span through several pages, with most of the data in the provided tables or charts, the authors should consider shrinking the results a bit, in favor of the provided data, to make it easily readable. Do not shrink it a lot, just omit things that are already clearly visible from the tables, or not that important. 

- I liked the gender discussion very much, it would be interesting to see future research in this direction.... this is just a note, you do not have to change anything in this paper.

The language and style are ok, but sometimes the sentences are not easy to comprehend, I suppose you are not a native speaker (neither am I) - if you could give your paper to a native speaker to proofread it, it would improve the quality and the overall impression a lot.

Author Response

Thank you very much. We appreciate the time and effort taken to provide feedback on our manuscript. We are grateful for the insightful comments and valuable suggestions for improving the article. Following are the responses to your comment.

I have just a few remarks which should be considered by the authors:

Point 1: The paper starts with a bit bold presumption: "The world of academia is currently facing a serious crisis, a crisis of engagement". While I personally and intimately agree, this statement should be supported by a citation (or, even better a few of them) since it is a basis for the whole research and this paper in the end. 

Author Response 1: Thank you very much for pointing this out. We have added the citation to support the statement.  

Point 2:  the authors calculated Cronbach's alpha for their questionnaire reliability, but the questions seem to be rather diverse, asking different things (e.g. q8 looks like a true/false question, and q12 and q13 are more "location" based), are you sure you can calculate Cronbach's alpha for the whole questionnaire?

Author Response 2:  All 13 questions listed in Table 2 used a five-point scale, ranging from strongly disagree (SD) to strongly agree (SA). We cannot calculate Cronbach’s alpha when we have unordered categorical data with more than three categories.  In such cases, we factor analysis is preferred.

Point 3: the authors discuss the "likeness" of the game (Vortex sorting game), which is rather simple and could be considered "too simple" or "boring" by some (or many?) of the students. Maybe some of the results are influenced by the nature of the game.

Author Response 3: We agree with the reviewer’s point of view that this game might be simple when compared to some other complex games used for learning and may have influenced the results. The outcome of this study indicates that this game managed to engage the students for this particular course as a very high number of students completed the activity. If the game is complex, it might switch off the students. On the other hand, some learners may find the temptation to keep playing the game too irresistible, so a clear expectation about learning time must be agreed upon with students.

Point 4: The results span through several pages, with most of the data in the provided tables or charts, the authors should consider shrinking the results a bit, in favor of the provided data, to make it easily readable. Do not shrink it a lot, just omit things that are already clearly visible from the tables, or not that important. 

Author Response 4:  We that all reported results are relevant because they reflect various aspects of the analyses, first focusing on the activities and then on the survey. However, after careful consideration, we have removed Table 4, which contains redundant information as the same is also provided in Figure 4. Other information is provided in the text now.

Point 5: I liked the gender discussion very much; it would be interesting to see future research in this direction.... this is just a note, you do not have to change anything in this paper.

Author Response 5: We intend to use different games with different features for both genders and explore the difference. We would also like to evaluate how the same game affects the learning of different genders.

Reviewer 2 Report

The topic is out of date for at least 5 years. Authors should have used AR or VR elements for gamifing the educational process. The Quiz and similar games (which can be found in various LMS) are not intended for Millenials.

The questionnaire is not correct because every question is suggestive. There are no negative statements to be used as control statements.

Author Response

Thank you very much. We appreciate the time and effort taken to provide feedback on our manuscript. We are grateful for the insightful comments and valuable suggestions for improving the article. Following are the responses to your comments.

Point 1: The topic is out of date for at least 5 years. Authors should have used AR or VR elements for gamifying the educational process. The Quiz and similar games (which can be found in various LMS) are not intended for Millennials.

Author’s Response 1: We disagree with the reviewer’s comment that this topic is out of date as a simple search on Scopus resulted in over 1000 articles, conference papers, and lecture notes published in the last three years. We agree that AR is a relatively new technology, but it has its own issues which make it hard to implement in the classroom. It requires a device with higher computation power, a high-speed internet connection, long battery life, and data storage capacity. Furthermore, there is a lack of tools or apps which can quickly be adopted for instruction design. Furthermore, what actually matters is the suitability of instruction design in the context irrespective of the age of technology. The use of the vortex game has been justified in the text as it requires minimal resources. However, we believe that Augmented Reality Game-Based Learning (ARGBL) will become increasingly relevant in Technology-Enhanced Learning as above mentioned issues are resolved.

Point 2: The questionnaire is not correct because every question is suggestive. There are no negative statements to be used as control statements.

Author’s Response 2: We agree with the reviewer’s comment and there is a long tradition of including items in questionnaires that are phrased both positively and negatively to minimize extreme response bias and acquiescent bias. Despite published concerns about acquiescence bias, there is little evidence that the common practice of including both positive and negatively worded items solves the problem. The research conducted by Sauro and Lewis [1] has found little evidence for these biases and reported that response bias effects are at best small and outweighed by the real effects of miscoding and misinterpreting by users. It should be noted that the participants are not native English speakers. Furthermore, including negative questions will make the questionnaire too lengthy, which could affect the completion rate. Also, we used a five-point scale, ranging from strongly disagree (SD) to strongly agree (SA), to capture the opinions of the participants on 13 statements given in Table 2. Therefore, the negative statements are reflected by the scale.

[1] Sauro, J., & Lewis, J. R. (2011, May). When designing usability questionnaires, does it hurt to be positive? In Proceedings of the SIGCHI conference on human factors in computing systems (pp. 2215-2224).

Reviewer 3 Report

Thank you for your paper on the effect of digital game-based learning on student engagement and motivation.  After reading your paper I do have a number of observations:
- gamification may be inspired by different digital game formats. In your study you selected the 'competition game" and in your methodology, this game type has been linked to temporal metrics. Your interpretation of the results must be seen in the context of these constraints;
- Schwabe & Göth, 2005, did define 6 criteria to create a useful digital gamification of learning material - criteria which can be found also in HCI studies - 

- student motivation and engagement has everything to do with realizing learning outcomes of the course and the program. In your paper this framing is missing.

- feedback is important in any educational process and as such you mentioned formative learning, but you didn't explore/elaborate on this in relationship to your 10 activities. In realizing the activities and as such addressing motivation and answering the survey questions - I do miss the exact questions per activity and why these questions have been formulated-the impact of feedback and connection to learning outcomes must have played a role. However, the course you selected seemed of rather low importance regarding the learning outcomes of the curriculum.

- regarding the variability of respondents you may question if you may compare these as results or do you need to normalize before. Could you explain why you didn't normalize the raw data?

- tables need more information in the annotation. Continue the same  coding of activities in your tables

- it is always of interest to mention the software tools you used. Of more importance is/are the methods/functions that you applied by means of the software.

in general, the text is very well readable. There are a minor number of sentences that needs rephrasing [lines: 40, 136, 144, 185, 265 (regarding previous evaluation scores?)

The bar charts may show by color use the immediate difference between game-based and 'traditional' [ TCG4 & TCG5]  didactic formats.

Author Response

Response to Reviewer 3 Comments

Thank you very much. We appreciate the time and effort taken to provide feedback on our manuscript. We are grateful for the insightful comments and valuable suggestions for improving the article. Following are the responses to your comments.

Point 1: gamification may be inspired by different digital game formats. In your study you selected the 'competition game" and in your methodology, this game type has been linked to temporal metrics. Your interpretation of the results must be seen in the context of these constraints;

Response 1: Thank you very much. We have provided a detailed discussion of the relationship between the two. Students are competing against each other and the parameter measuring the performance is the time taken to complete the activities. Competition motivates the students to do well and reduces the time to complete the activities which is only possible if students grasp the concepts. Thus, a lower score means that the student is motivated to do better.

Point 2:  Schwabe & Göth, 2005, did define 6 criteria to create a useful digital gamification of learning material - criteria which can be found also in HCI studies –

Response 2: Thank you very much for pointing this out. We have provided the rationale for selecting this game based on the structure elements defined in the recommended publication. 

Point 3:  student motivation and engagement has everything to do with realizing learning outcomes of the course and the program. In your paper this framing is missing.

Response 3: A student's motivation is what propels them to take action. The visible activity or proof of that motivation is engagement. Engagement requires motivation, yet successful engagement may also encourage students to remain motivated in the future. Meaningful engagement fosters deeper learning. These activities supplement the current instruction design which is aligned with the course learning outcomes. It means completing these activities will help students in achieving the learning outcomes. In addition to being a goal in and of itself, student engagement in learning also serves as a vehicle for students to achieve successful academic outcomes.

We can also measure how helpful these activities were in achieving the learning outcome. We do have data in the form of assessment grades as this course has six assessments, each of them covering one or two topics. These assessment grades can be compared against the activity scores to see if any correlation exists. This can provide insight into whether activity scores have any impact on assessment grades. These grades provide an indication that how successful the student was achieving the learning outcomes. But the literature does not recommend using assessment grades to assess the students learning as a result of these activities. This issue can be addressed in future work.  

Point 4:  feedback is important in any educational process and as such you mentioned formative learning, but you didn't explore/elaborate on this in relationship to your 10 activities. In realizing the activities and as such addressing motivation and answering the survey questions - I do miss the exact questions per activity and why these questions have been formulated-the impact of feedback and connection to learning outcomes must have played a role. However, the course you selected seemed of rather low importance regarding the learning outcomes of the curriculum.

Response 4: We have based our study on the hypothesis that intrinsic motivation creates deeper engagement which leads to deeper learning. Our focus is to motivate the student to engage with the learning material. Instrumental students pay more attention to the activities which can help them to improve their grades. In this context, students are least bothered to pay attention to such a course which does not help them in improving their GPA. It is a very challenging task to motivate the students to engage with the material in such a course and this was the rationale behind the selection of this course for implementing game-based learning. This weakness pointed out is actually the main strength of our study. If this instruction strategy proves successful in motivating and engaging students, then its success is guaranteed in another high-importance course.

Point 5: regarding the variability of respondents, you may question if you may compare these as results or do you need to normalize before. Could you explain why you didn't normalize the raw data?

Response 6: There are two reasons for not normalizing the raw data. First, both GBQA and MBQA results are in different formats. GBQA results are in seconds whereas the MBQA results are in percentile format. An attempt to normalize the data provides results that are not suitable for the reader. Also, different activities contain different numbers of questions, and the time to complete the activities has different ranges making it a less attractive option for normalizing. Furthermore, activities are orthogonal to each other and we compare the results of the same activity for different groups and normalizing the data does not provide any added value.

Point 6:  tables need more information in the annotation. Continue the same coding of activities in your tables.

Response 6: Thank you very much for pointing this out. We have provided more information in the annotation and consistent coding is provided in the revised manuscript.

Point 7: it is always of interest to mention the software tools you used. Of more importance is/are the methods/functions that you applied by means of the software.

Response 7: We used R-software to perform the statistical analysis of the results. First, we performed a univariate analysis including the descriptive statistics of the survey results. We then performed a bivariate analysis including the hypothesis tests for comparing the results of different groups in the activities.

Point 8: in general, the text is very well-readable. There are a minor number of sentences that needs rephrasing [lines: 40, 136, 144, 185, 265 (regarding previous evaluation scores?)

Response 8: Thank you very much for pointing this out. We have rephrased the sentences in the revised manuscript.

Point 9: The bar charts may show by color use the immediate difference between game-based and 'traditional' [ TCG4 & TCG5] didactic formats.

Response 9: We agree with the reviewer’s comment that the graph presents trends and patterns in a better way, but they do not present more information such as min./max. and SD. Therefore, in this table is more suitable for conveying the required information to the readers.

Round 2

Reviewer 3 Report

Thank you for responding to the review comments, it did make your choices regarding the course to support by gamification better understandable.

Author Response

Point 1: Thank you for responding to the review comments, it did make your choices regarding the course to support by gamification better understandable.

Response 1: Thank you very much. We appreciate the time and effort taken to provide feedback on our manuscript. We are grateful for the insightful comments and valuable suggestions for improving the article.